# Immunobiology and Application of *Aloe vera*-Based Scaffolds in Tissue Engineering

**DOI:** 10.3390/ijms22041708

**Published:** 2021-02-08

**Authors:** Saeedeh Darzi, Kallyanashis Paul, Shanilka Leitan, Jerome A. Werkmeister, Shayanti Mukherjee

**Affiliations:** 1The Ritchie Centre, Hudson Institute of Medical Research, Clayton 3168, Australia; saeedeh.darzi@hudson.org.au (S.D.); kallyanashis.paul@monash.edu (K.P.); slei0003@student.monash.edu (S.L.); jerome.werkmeister@hudson.org.au (J.A.W.); 2Department of Obstetrics and Gynaecology, Monash University, Clayton 3168, Australia

**Keywords:** AV, biomaterials, regenerative medicine, foreign body response, macromolecules, anti-inflammatory material, bioink, tissue engineering

## Abstract

Aloe vera (AV), a succulent plant belonging to the Liliaceae family, has been widely used for biomedical and pharmaceutical application. Its popularity stems from several of its bioactive components that have anti-oxidant, anti-microbial, anti-inflammatory and even immunomodulatory effects. Given such unique multi-modal biological impact, AV has been considered as a biomaterial for regenerative medicine and tissue engineering applications, where tissue repair and neo-angiogenesis are vital. This review outlines the growing scientific evidence that demonstrates the advantage of AV as tissue engineering scaffolds. We particularly highlight the recent advances in the application of AV-based scaffolds. From a tissue engineering perspective, it is pivotal that the implanted scaffolds strike an appropriate foreign body response to be well-accepted in the body without complications. Herein, we highlight the key cellular processes that regulate the foreign body response to implanted scaffolds and underline the immunomodulatory effects incurred by AV on the innate and adaptive system. Given that AV has several beneficial components, we discuss the importance of delving deeper into uncovering its action mechanism and thereby improving material design strategies for better tissue engineering constructs for biomedical applications.

## 1. Introduction

The *Aloe vera* (AV) plant has been used for centuries for its health, beauty, medicinal and healing properties. Although its first origin is uncertain, it derives its name from ancient Arabic nomenclature “Alloeh; meaning shining” and “Vera; meaning genuine”. AV grows wild in tropical, semi-tropical and arid climates around the world [1,2]. Since discovery, AV has been considered as a natural healer. The beneficial effects of AV have been reported in several areas, including anti-bacterial, anti-oxidant, anti-inflammatory and wound healing. Today, AV is widely used in food, cosmetics and healing creams and its general medicinal properties have had broad applications in skin injuries, anti-cancer therapeutics, diabetes and dental areas. More recently, we have demonstrated even broader applications in regenerative medicine in the form of a biomaterial for stem cell delivery and tissue engineering applications [3,4].

The evergreen perennial plant of the genus Aloe comes with 420 species in various synonyms [3,4,5]. Among them, AV (scientific name: *Aloe barbadensis* Miller) found a more widespread application than any other plant. The plant has green spiky succulent and fleshy leaves that exudate a bitter juice and a transparent, slippery mucilage/gel. Under strongly cutinised epidermis, the pericyclic cells produce the former exudate and centrally located thin-walled tubular cells in parenchyma are produced later [4,6]. About 99% of the total exudates are water and the remaining 1% contains all the bioactive components, including phytochemicals, vitamins, minerals, enzymes, anthraquinones and different lectins, totalling more than 75 known substances [7].

The growing therapeutic appreciation for AV stems from emerging scientific evidence in several diseases, such as rheumatism, Parkinson’s disease, asthma, diabetes, cancer, burns and infections [8]. This has fuelled the use of AV and its bioactive components, to be incorporated with biomaterials as well as cellular therapy to enhance the repair and regeneration of damaged tissues. This review describes the AV plant, its constituents, and their biological properties. We highlight AV’s importance and advantage in the form of a biomaterial, and in promoting regeneration for tissue engineering applications. Furthermore, this review focuses on the critical elements of the immune system and the mechanisms that drive the anti-inflammatory and healing effects of AV. Finally, we discuss the clinical and foreign body response considerations that would directly impact AV-based implantable scaffolds’ design and their application in the biomedical field.

## 2. Bioactive Components of *Aloe vera*

According to the World Health Organization (WHO), AV is the most bioactive plant among all the 420 aloe species [3]. The phytochemicals from AV possess pharmacological activities [9,10] as listed in Table 1.

These diverse bioactive natural agents are useful and safer against many chronic diseases like cancer. AV can infiltrate into the tissue and increase the transportation of the cell nutrients, enzymes and oxygen content through the blood circulation [6]. On a dry matter basis, AV gel contains polysaccharides (55%), sugar (17%), minerals (16%), proteins (7%), lipids (4%) and phenolic compounds (1%) [2]. The bioactive constituents of AV are divided into two major types; namely, nutritive and non-nutritive constituents. The nutritive constituents include carbohydrates, vitamins, enzymes, minerals and trace elements, protein and amino acids, whereas the non-nutritive constituents include phenolic compounds, organic compounds, phytosterols and other compounds [7,11]. The bioactive constituents of AV include several amino acids required by our body, including seven essential amino acids (Figure 1).

### 2.1. Polysaccharides

AV gel mainly contains over 99% water. Besides water, other material components account for about 0.5–1% of AV gel [13]. Of these, the carbohydrate-containing fraction (~0.25% of AV gel) is the most significant fraction and may vary from 25–50% of solid precipitates [14]. The carbohydrate fraction includes monosaccharides (mannose, fructose and galactose), polysaccharides, namely long-chain linear gluco-mannans, β (1,4)-linked acetylmannan presenting glucose and mannose subunits, galactan, galactogalacturan, free sugars (glucose), fibres, cellulose and xylan [14]. Although not fully understood, the structure of carbohydrates in AV gel has been demonstrated by linkage and optical rotation analysis. The oligosaccharides produced by enzymatic or acid hydrolysis contain β-glucomannan backbone with a Man/Glc ratio ~15:1 [9]. The detailed composition analysis is reported using the vacuum filtrated bulk water soluble polysaccharide (BSW) suggests the monosaccharaide compositional ratio of Man/Glc/Gal/GalA/Fuc/Ara/Xyl is 120:9:6:3:2:2:1 [9,15]. Acetylated mannan or acemannan is the primary carbohydrate fraction present in AV gel with beneficial medicinal properties [16]. FTIR analysis of AV gel shows strong bands within the range of 1078–1036 cm^−1^ indicating the presence of mannose and glucose sugars, the primary monomers. The transmittance spectrum around 1740 and 1248 cm^−1^ may be attributed to the presence of C=O and C−O−C stretches in acetyl groups corresponding to the bioactive acetylated acemannan [16]. Since polysaccharide content is the principal constituent of AV gel, they have been widely explored for their distinctive bioactive roles. AV polysaccharides are immune stimulants that display adjuvant activity on specific antibody production and enhance the release of interleukin-1 (IL-1), interleukin-6 (IL-6), tumour necrosis factor-α (TNF-α) and interferon-γ (INF-γ) [9]. Bioactive acemannan controls immunomodulation and anti-tumour properties of the gel [17,18]. It mainly induces macrophage cytokine production, nitric oxide (NO) release, surface molecular expression and cell morphological alteration in a mouse macrophage cell line [15,17].

### 2.2. Vitamins, Enzymes and Minerals

AV gel is widely known as an anti-oxidant agent due to the presence of vitamin C (ascorbic acid), carotenoids, vitamin E (tocopherols), vitamin B1 (thiamine), vitamin B2 (riboflavin), vitamin B6, niacin and folic acid. AV gel contains at least six enzymes; reportedly cellulose, carboxypeptidase, amylase, bradykinase, oxidase and catalase, that help digestion and nutrient absorption out of food decomposition (fats, sugars) [7,19]. Additionally, minerals and trace elements such as magnesium (Mg), calcium, (Ca), iron (Fe), copper (Cu), zinc (Zn), potassium (K), chloride (Cl), manganese (Mn) and chromium (Cr) are also present in AV gel [7,19]. The presence of trace elements such as Mg contributes to the anti-allergic properties of AV gel [11]. The essential minerals such as Fe, P, Cu, Mn and Ni are crucial for good health. They have been investigated for their role as anti-oxidants and functioning of different enzymes in various metabolic pathways. Some other trace elements, such as lead, Pb, boron, B, strontium, Sr, aluminium, Al and cadmium, Cd are reported as toxic elements [20].

### 2.3. Phenolic Compounds

The non-nutritive constituents of AV include phenolic compounds, organic acids, phytosterols and other compounds, namely aliphatic hydrocarbons, long-chain esters and volatile components such as aldehydes and ketones [7]. The phenolic compounds are the lead bioactive compounds of AV as potential anti-cancer agents [21]. These polyphenols are known as bioactive anthraquinones, an anti-oxidant agent. They are mostly present under the cutinised AV leaf and in lesser content in its roots and gel. The spectroscopic characterisation using ^1^H NMR (400 MHz) and ^13^C NMR (100 MHz) shows the presence of aloesaponarin-I, deoxyerythrolaccin, lactic acid D methyl ester, aloesaponarin-II, aloesaponol-I and aloe-emodin; 1,8-dihydroxy-3-(hydroxymethyl) [10]. Components such as anthraquinones help absorb food from the gut and show considerable anti-microbial, analgesic and anti-viral potential when present in lower amounts [2]. Anthraquinones are potential anti-microbial agents of AV that act similar to tetracycline and block the bacteria protein synthesis, therefore inhibiting bacterial growth, including *Escherichia coli*, Staphylococci, *Enterococcus faecalis* and resistant *Helicobacter pylori* [13]. Moreover, the anthraquinone component inhibits the activity of enzymes such as nicotinamide adenine dinucleotide and mitochondrial succinate oxidase, responsible for the electronic transfer from the bacterial respiratory chain. Dehydrogenation in the bacterium can damage the integrity of the membrane, causing loss of the cytoplasm. In vitro studies on the anti-microbial effects of AV gel showed the effectiveness of AV gel to inhibit Gram-negative and positive bacteria through the induction of leukocyte phagocytosis [22]. Additionally, some pharmaceutical industries use aloin for the preparation of diacerein, a drug used for treating osteoarthritis. As potential anti-cancer agents, aloe-emodin, aloin, chrysophanol, aloesaponarin I, aloesaponarin II and aloesin have been tested for various cancers such as hepatoma carcinoma, cervical cancer, breast cancer, acute lymphoblastic leukaemia, colorectal cancer, ovarian cancer and oral carcinoma. Most of these studies showed their potential to increase apoptosis, multinucleate cells, cell death, DNA fragmentation, caspase 3,8,9 and decrease mitochondrial membrane potential, cell migration and invasion [21].

### 2.4. Processing of AV

The evergreen perennial plant AV is processed in an industrial setting and extracted as liquid juice or in powder form. The production process involves filleting, crushing, grinding or pressing the entire AV leaf [23]. Thus, the AV juice follows filtration, deaeration, sterilisation, and involves hot or cold processing following preservatives and stabiliser addition. For powder form, AV leaf is processed to obtain the AV gel fillet, which undergoes washing and is then placed in a humidified chamber at an appropriate temperature [23]. Hot air is passed over the fillet for further drying, followed by grinding to produce the final powdered form. In general, bioactivity, colour and flavour are restored through freeze drying rather than hot air drying [23]. For applications in tissue engineering, AV is best used in powder form, and is often blended with degradable polymers.

## 3. Biological Applications of *Aloe vera* and Its Components

### 3.1. Anti-Cancer

As medications for cancer chemotherapy have high toxicity toward non-cancerous cells and tissues, phytochemicals or bioactive compounds from natural sources have been investigated to develop anti-cancer agents such as paclitaxel, docetaxel, etoposide, topotecan and irinotecan [21]. Of these, AV components have shown promising anti-cancer activities: anti-proliferation [24], cell cycle inhibition [25], induce apoptosis [26,27], anti-inflammation [28], upregulation of tumour suppressor genes [29], down-regulation of oncogenes, regulation of hormonal levels, growth factor regulation, and suppression of invasion and metastasis [2]. AV has been reported to have many biological components which have anti-oxidant, anti-bacteria/viral, anti-inflammatory and immunomodulatory properties [6] (Figure 2). Inhibitory effect of acemannan and its chemo-preventive action by inhibition of B[a]P absorption has been known for a long time [30]. In the last decade, the beneficial effects of aloe-emodin in inducing cancer cell death through apoptotic pathways has emerged [31,32,33]. Its role in regulating mitochondrial proteins and factors and thus impacting the cancerous microenvironment is now well understood. Aloe-emodin is now known to exhibit anti-cancer activity against established leukaemia [29], gastric [34], colon [35], glial [25] and lung [26] cancer cell lines.

### 3.2. Anti-Oxidant

Oxidative stress is an imbalance between disposal and generation of reactive oxygen and nitrogen species (ROS/RNS). These reactive species are harmful to cells and are involved in inflammation [36]. AV has potent anti-oxidative capacity that can scavenge oxygen radicals and inhibit iron oxidation [37]. Anti-oxidant activity of AV is related to a substantial content of phenolic components [38]. About 18 phenolic constituents have been identified in AV, including catechin, sinapic acid, quercitrin, gentisic acid and epicatechin [39]. In vivo assessments of AV anti-oxidant activity have shown that AV gel not only protects against the 8G radioactive damage but also delays the onset and severity of radiation sickness in affected mice [40]. The ethanolic extract of AV gel has a significant anti-oxidative activity, reducing thiobarbituric acid reactive substances, hydroperoxides, superoxide dismutase, catalase and glutathione peroxidase in streptozotocin (STZ)-induced diabetic rats [41].

### 3.3. Anti-Viral

Acemannan is associated with mediation of anti-viral effects of AV. It directly engages with cells of the innate and adaptive immune system and influences antigen presentation, T cell activity and antibody production [42]. A preliminary clinical trial study showed the efficacy of AV on HIV patients’ immune systems by a slight increase in CD4 count after daily usage of 30–40 mL of AV [43]. AV anthraquinone derivates such as aloe-emodin also exhibit anti-viral activity. It reduces the virus-induced cytopathic effect and inhibits replication of influenza A [42]. The effect of AV on the influenza virus suggested an indirect impact of AV on virus replication through upregulation of galectin 3 [42]. Galectin 3 increases the expression of anti-viral cytokine IFN genes, including IFN-β and IFN-γ [42]. Aloe-emodin is also effective against several virulent infectious agents such as herpes and pseudorabies viruses [44].

### 3.4. Anti-Microbial

Given the health burden of microbial infection on public health, the anti-microbial property of AV presents enormous potential. AV gel works against both Gram-positive and negative bacteria [45]. In particular, anthraquinones are structurally similar to antibiotic tetracycline. As a result, they are effective in overcoming bacterial growth through inhibition of the translation process. In vitro studies have shown that AV gel prevents bacteria from adhering to human epithelial cells, thus stopping the infection process [46]. Drug resistance of bacteria is a critical global health issue. Interestingly, AV can combat resistant strains of *H. pylori* and prevent gastric infections [47]. AV components have also been shown to act against fungal infections [48]. Thus, owing to its wide range of anti-microbial action, AV and its bioactive components present a promising opportunity to incorporate these effects in tissue engineering scaffolds.

### 3.5. Skin

AV has found significant popularity in dermatology, for a wide range of reasons, ranging from hydration, UV exposure, burns, as well as ageing. The anti-oxidative properties of AV gel are known to combat the oxidising biomolecules and free-radicals that harm the skin from radiation damage [2]. It further induces fibroblasts to produce extracellular matrix, such as collagen and elastin, and hence is a commonly used additive in anti-ageing creams. AV components, such as amino acids and zinc, render cohesive effects on epidermal cells, thereby inducing softening [19]. Additionally, AV has a significant impact on modulating keratinocytes [49] and cytokine milieu of tissues and has found application in wound healing [2]. This has inspired the design of several biomaterials for burn and wound healing to incorporate AV as a strategy [49].

## 4. Application of *Aloe vera* in Tissue Engineering

Tissue engineering (TE) is a thriving field of regenerative medicine that aims to develop biocompatible substitutes that restore or improve damaged tissue function. TE involves a combination of materials, cells and bioactive molecules to form a therapy to enhance people’s quality of life and address critical health issues [50]. TE primarily depends on scaffolds mimicking the injured/damaged site with appropriate vasculature of the tissue/organ that needs to be repaired or even replaced [51]. Given the therapeutic advantages of AV, designing and engineering composite biomimetic scaffolds incorporating AV is a desirable approach. Natural biomaterials such as AV provide the biomimetic cues that enable them to fit the normal microenvironment of tissues better, thus promoting desirable cellular responses, biocompatibility and degradability [52]. The AV gel/powder format is a house of bioactive phytochemicals and minerals that can actively enhance the TE construct functionality through its anti-inflammatory, anti-bacterial, antiseptic and regenerative properties. Given that AV-based scaffold can be fabricated using different techniques, including electrospun nanofibres, hydrogels and 3D printing, their potential in TE is indeed vast [53].

### 4.1. Mesh and Mats

Over the last decade, AV has found widespread applications in TE in various forms like hydrogel [54], bioprinted constructs [55] or drug delivery applications [1,56]. In hydrogel format, the AV is usually blended with more commonly used natural biomaterials like chitosan, gelatin, collagen [57], alginate [58], silk [59] and nano cellulose [60]. AV in dry powder form or nano cellulose fibre form [61] is mixed to physically entrap or blend into the polymer matrix to form an injectable composite hydrogel or lyophilised polymeric scaffold using freeze-drying. Combining AV to create meshes or mats has been recently explored in skin regeneration for burn patients, bone tissue engineering, corneal endothelial cell regeneration and cardiac tissue regeneration [62]. Recently, nanotechnology using electrospinning of AV with various synthetic polymers like PLLA [63] and PCL [64] has generated nanofibrous polymeric mats [6]. The blended electrospun constructs have prolonged retention of AV and sustained bioactivity in TE applications that accelerate wound healing, endogenous cellular proliferation, anti-microbial functionalities and aids immunosuppression and control release to cancer-like diseases.

In the form of TE constructs, AV controls physiochemical properties including tensile strength, elongation [65] and biodegradation and biological properties to various cells types like fibroblasts, endometrial mesenchymal stem cells (eMSCs) or macrophage polarisation in their attachment and proliferation [65]. The AV/synthetic PLGA, PCL polymers have shown that low concentrations of AV increases the tensile strength [66], whereas AV/natural biomaterials comprising chitosan and gelatin showed excellent porosity in hydrogel format but adding AV decreased the mechanical strength of the films. Considering the beneficial effects of synthetic and natural biomaterials, the novel composite scaffolds blending both with AV can potentially overcome the limitations and can be a potential hope for future TE constructs [3]. Due to the healing abilities of AV, its use within biomaterial engineering has gained significant interest. Yet, the application of AV in the form of a mesh or mat, such as electrospun or 3D printed scaffolds, has been sparsely studied. Nanofibre electrospun meshes are an extremely viable AV delivery method as nanofibres allow for gas exchange and waste excretion where other methods would inhibit these functions [66]. AV in nanofibrous scaffolds is likely to provide a suitable environment for cell viability and proliferation. It can also enhance scaffold hydrophilicity and protein absorption capacity. Studies show that AV improves fibroblast proliferation and reduced cell cytotoxicity in hybrid composites of NFs/ZnO/Alv [67] and COL-CS-AV [67]. In another study, PLGA nanofibrous scaffold, which contained epidermal growth factor and AV promoted fibroblast proliferation and significantly improved wound closure and re-epithelisation in an in vivo full-thickness wound healing assay in diabetic mice [6]. PCL nanofibres, which contain 10% AV, enhanced cell proliferation. Cellular movement using CMFDA (5-chloromethylfluorescein diacetate)-labelled cells showed a significant increase in fibroblast movement, collagen secretion and F-actin expression in 10% AV blended PCL scaffolds compared to PCL controls [64]. AV function was further investigated by incorporating it into nanofibrous scaffolds that mimic the Extracelluar Matrix ECM [68]. These scaffolds were made of co-polymers such as PLACL. The addition of AV to PLACL meshes resulted in an increased capability to endure strain due to an increased elastic potential provided by AV. Examinations also showed that cells attach and proliferate more successfully in meshes that were infused with AV [69].

Essentially, the addition of AV to PCL or PLACL scaffolds provided a stable environment for tissue growth whilst releasing essential growth factors for tissue regeneration [65]. As shown in Figure 3, meshes infused with AV at both 5% and 10% have significantly increased host cell interactions. Given this, it is clear that incorporating AV provides vital support, enhancing cellular interactions and aiding in healing processes.

Our group has shown that AV in the form of a hydrogel used as bioink increased biocompatibility and healing by increasing collagen deposition and host cell infiltration [55]. Yet, the impact of blending AV into the polymeric meshes has not yet been fully explored. In particular, the in vivo immunological implications remain unknown. From a translational perspective, it is important to not only design effective scaffolds for tissue repair, but also understand the mechanisms that drive the healing process and immunological considerations that may impact its degradation.

### 4.2. Hydrogels

In contrast to electrospun TE constructs, application of AV in the form of hydrogel has been widely explored for multiple biomedical applications. Combining hydrogels with electrospun meshes can augment the tissue integration properties of the overall TE construct. We have reported bioprinting using AV hydrogel, to promote the integration of 3D printed scaffolds as a novel concept. Our previous reports have shown that bioactive AV functionalises bio inert alginate to instantly form a thixotropic in situ hydrogel in the absence of Ca^2+^ ions [55]. This AV alginate hydrogel can be used as a bioink to bioprint eMSCs for gynaecological applications such as pelvic organ prolapse (POP) [55]. A recent breakthrough study from our group has shown that AV-based injectable hydrogel can treat vaginal injury incurred during childbirth [55]. The treatment is further enhanced through a tissue engineering strategy incorporating eMSCs. This implies that an injectable treatment soon after birth may effectively reverse birth trauma, as shown in Figure 4. Given that over 60% of childbirths are traumatic [54], such minimally invasive treatment using AV is likely to significantly alleviate maternal birth injury. Furthermore, AV hydrogel has desirable properties, such as biodegradation, nutrient transportation for better cell proliferation, anti-microbial properties, prolonged retention of stem cells and repair of injured tissue [54,55].

Repair and regeneration of damaged tissue involves the formation of new connective tissue and angiogenesis during healing [53]. AV is well known for its wound healing properties, although the underlying mechanisms are not fully understood. Recent studies have shown that AV increases the collagen content in granulation tissue with lower type I/type III collagen ratios in AV-treated groups than the untreated wound healing animal model [70]. Compared to standard healing agents, such as 1% silver sulfadiazine and salicylic acid cream, AV has substantially quicker healing effects on full-thickness wounds on second-degree burns in guinea pigs [71]. Delivering AV in an examination glove to 30 working females suffering from occupational dry skin improved their skin integrity and decreased erythema and fine wrinkles [72]. Several other studies have shown the healing effects of AV on psoriasis, mouth sores, ulcers, diabetes, herpes, bedsores and burn wounds [6]. Although not fully understood, evidence suggests that the glucomannan compound in AV, rich in mannose, affects fibroblast cellular activity and proliferation [73]. The interaction of AV components with growth factor receptors consequently increases the collagen synthesis and imparts healing. Furthermore, AV gel increases cell surface expression of β1-, α6-, β4-integrin and E-cadherin in a keratinocyte cell line (HPEK) that drives cell migration and wound healing [74]. AV also increases hyaluronic acid and dermatan sulphate in the wound area. Hyaluronic acid, a key component of ECM structure, has an essential role in the wound healing process [75]. A 10% AV gel reduced the vascularity and the number of mast cells, whereas it increased the fibroblast cells in the synovial fluid of a synovial air pouch rat model [76]. The enhanced wound healing properties of AV may be attributed to its bioactive components, such as vitamin C, E and amino acids. In general, vitamin C increases collagen production, whereas vitamin E is a potent anti-oxidant promoting wound healing [38]. Evidence shows that emodin, one of the derivatives of anthraquinones, can promote tissue regeneration in a rat model [77]. Moreover, AV mucilage increases collagen on the wound site with enhanced transversal connections [71]. Besides wound healing effects, AV promotes repair through epithelial cell migration, collagen maturation and inflammation [6].

Furthermore, AV is also known to promote angiogenesis in open wounds in type 2 diabetic rats through oral administration with increased TGF-β1 and VEGF production in cells just 4 days post-injury in AV-treated animals [78]. Topical application of AV-derived eye drops was highly effective in treating alkali-burned cornea, hastening re-epithelialisation [79]. Beta-sitosterol content of AV showed a potent angiogenic activity while stimulating neo-vascularisation [38]. The rich supply of nutrients and natural bio components found in AV present several opportunities for wound healing. For example, hydrolysing enzymes in AV work to reduce pain and inflammation, while amylase enzymes act to degrade necrotic tissue [3,6,80,81,82]. Aloctin-A causes macrophages to phagocytose dead tissue [64,83]. Furthermore, amino acids aid in tissue growth and regeneration, whilst vitamins [84] are consumed to provide anti-oxidant effects within the host tissue [64].

## 5. Foreign Body Response to Implanted Biomaterials

The immune system is responsible for eliminating any foreign threat and maintaining the homeostasis of the body. There are several cytokines and chemokines that enable a competent signalling system along with the cells of the immune system. Biomaterials and TE constructs are considered foreign, and following their implantation, the immune system is immediately activated [85] to trigger a foreign body response (FBR). This activation involves a cascade of events, as shown in Figure 5, initiated by protein adsorption and resulting in a provisional matrix around the biomaterial, leading to platelet aggregation and cellular recruitment such as neutrophil, mast cells and monocytes [53]. Acute and chronic phases of FBR follow the formation of the provisional matrix. The intensity of FBR is primarily determined by the composition and design properties of implanted scaffold, and the extent of the damage incurred [66]. Irrespective of the macromolecules or polymer used, it is pivotal to understand the interactions between the immune cells and cytokines with the implanted biomaterials. From a TE perspective, a hallmark of a successful implant is determined by its FBR immunobiology at the host tissue interface. Implants that lead to undesirable inflammatory response are likely to fail and have serious consequences. For instance, non-degradable polypropylene meshes frequently used in transvaginal reconstructive surgeries incurred unfavourable FBR that led to serious health complications due to non-integration with host tissue [49,67]. As a result, these materials were withdrawn from the market and ultimately banned in many countries.

Acute immune response to implanted scaffolds is indicated by neutrophil recruitment and activation, as shown in Figure 5. Activated neutrophils attempt to make an oxidative environment by secreting proteolytic enzymes and reactive oxygen species (ROS). Following activation, neutrophils produce potent chemokines CCL2 and CCL4, which recruit monocytes, immature dendritic cells, macrophages and lymphocytes. By lymphocyte and macrophage accumulation, the acute phase of FBR is resolved. The neutrophils subsequently undergo apoptosis and are cleared from the implantation site. The monocytes that were recruited differentiate into M1 macrophages producing inflammatory cytokines such as IL1β and TNFα. Depending on the cytokine milieu and the type of biomaterial, the monocytes may polarise to either M1 or M2 macrophages, as shown in Figure 6, which have distinctive roles [53,86]. Using ROS enzymes, M1 macrophages try to digest the biomaterial [87]. M1 macrophages differentiate to M2 macrophages with reduced degradative ability and produce anti-inflammatory cytokine IL10, mannose receptor and arginase I. M2 macrophages induce the proliferation and migration of fibroblast cells leading to ECM synthesis and tissue regeneration [88]. The timely transition of M1 to M2 phenotype is the critical aspect of FBR, identifying the fate of implanted biomaterial [53,86,89].

The continuous presence of M1 at the implantation site results in chronic inflammation, ultimately leading to fibrosis and scar formation. M1 to M2 transition is highly dependent on biomaterial properties, tissue microenvironment and the associated inducing factors or regenerative cells. Constant presence and activity of macrophages at the site of implantation results in the formation of foreign body giant cells (FBGC), which is the hallmark of the chronic phase of FBR. FBGC results from several macrophages’ fusion, as shown in Figure 7, attempting to phagocyte a large particle [53]. The adaptive immune system also plays a critical role in FBR, mainly through T lymphocyte activity. T-cells can adhere to biomaterial surfaces in vitro and enhance the fusion into FBGCs through paracrine actions of secreted cytokines [90]. Moreover, growing evidence suggests that T lymphocytes play a critical role in regulating the host response to biomaterials through their interaction with macrophages and fibroblasts [87].

Anti-inflammatory and anti-fibrotic M2 macrophages interplay with regulatory T cells and have an essential role in inhibiting inflammation and restoring tissue homeostasis. Regulatory T cells (Tregs) induce anti-inflammatory M2 macrophages by secreting immunomodulatory cytokines like TGFβ and IL10 [91]. This synergic loop between M2 macrophages and Tregs attenuates the tissue injury and promotes regeneration. The vast number of studies around the interaction between the immune system and the implanted biomaterial have identified the critical role of macrophages. Therefore, the emerging TE research now focuses on designing immune-friendly biomaterials that can promote a desirable interaction with immune cells and ultimately aid repair and regeneration. While many studies have revealed the role of macrophages, the breadth of T cell activity in FBR and cellular immune response remains unclear. Given the tight interaction between the innate and adaptive immune system, it is also critical to uncover the role of adaptive immune cells to design safe biomaterial with long-term efficacy.

Given the anti-inflammatory effect of AV, it is likely that its immunomodulatory effect stems from the interaction of its components with molecules and cells of the innate and adaptive system. Therefore, AV has been considered a potential immunomodulatory additive for TE construct design that may aid in controlling FBR to implanted biomaterials.

## 6. Immune Response to AV-Based Scaffolds

AV is one of the well-known remedies for inflammation and wound healings. AV gel is an anti-inflammatory agent commonly used to combat pain and swelling. Such an effect is driven by the modulatory and stimulatory impact of AV on immune cells. Particularly, AV inhibits inflammatory cytokine IL1β, IL6, IL8 and TNF production in human immune cells [92]. Moreover, crude AV gel modulates nitric oxide release, surface molecules expression, as well as phagocyte activity in macrophages [38]. In vitro reports show that AV inhibits reactive oxygen metabolites and prostaglandin E2 (PGE2) production in CaCO_2_ epithelial cells [28]. More recently, AV vaccine has been shown to have stimulatory effects on the cellular and humoral immune system [93,94]. For instance, the oral administration of AV significantly enhanced CD4+ and CD8+T cells and IgG production in the rabbit’s blood injected with AV vaccine [93]. Additionally, daily treatment of mice for 14 and 21 days with 50 or 150 µL of AV gel resulted in enhanced chemotactic activity and the more robust response of their splenic lymphocytes to mitogen PHA [95]. Acemannan induces the maturation of immature dendritic cells, which was supported by increased allogeneic mixed lymphocyte reaction (MLR) and IL-12 production [96]. Acemannan also affects the macrophages in the immune system. The effect of acemannan on the macrophage cell line revealed that the mixture of acemannan and IFN γ stimulate cytokine production, NO synthesis and surface molecular expression in the RAW264.7 macrophage cell line [97]. Recent evidence shows that AV gel directly suppresses allergic responses through direct inhibition of type 2 helper T cells (Th2) and reduction of pro-inflammatory cytokines such as IL4, IL-5, IL-13, as well as histamine, mast cell protease-1 (MCP-1) and immunoglobulin IgE [18]. Interestingly, AV also increased the production of IL-10, an anti-inflammatory cytokine, and blocked the granulation of mast cells [18].

Anthraquinones, phenolic compounds of aloe leaves, have proven to be effective on the immune system. Comparing the anti-inflammatory effects of aloin and aloe-emodin (AE) with other polyphenols in AV demonstrated that aloe-emodin and aloin dose-dependently inhibited inducible nitric oxide synthase (iNOS) mRNA expression and nitric oxide (NO) production (5-4 microM). The levels of cyclooxygenase-2 (COX-2) mRNA and prostaglandin E2 (PGE2) production were suppressed by 40 microM aloe-emodin [98]. The other signalling pathway that might be affected by aloe-emodin is the p38 mitogen-activated protein kinase (MAPK) pathway. It has been shown that aloe-emodin reduces oxidative stress, inhibits inflammatory cytokine secretion and MAPK signalling pathway (essential kinases involved in inflammatory response) in a lung injury rat model [99]. AE also inhibits NFB/IRF5/STAT1 and IRF4/STAT6 signalling pathways, respectively regulating the LPS/IFN and IL4 responsive genes. This inhibitory effect modulates macrophage phagocytosis and migration as well as NO production [100]. The other component of AV that has shown an immunomodulatory impact is alprogen. Alprogen inhibits calcium influx into mast cells and as a result, inhibits the antigen-antibody mediated release of histamine and leukotrienes from mast cells [101].

Given its immunomodulatory effects, AV has been considered and investigated as a natural regenerative material in TE constructs reducing the inflammatory response. For instance, application of aloe-vera/chitosan nanohydrogel in a wound healing rat model showed an increased number of M2 (CD163+) macrophages 3, 7 and 14 days post-treatment and reduced iNOS expressing macrophages three days after treatment [102]. Intradermally injected *Aloe vera* hydrogel loaded with adipose-derived stem cells has significantly decreased the TGB-B1 and IL-1β at 7 days post-injury in a burn wound healing rat model [103].

We have shown that AV-based hydrogel significantly reduces the pro-inflammatory CCR7+ macrophages and increases CD206+ anti-inflammatory macrophages following surgical biomaterial implantation of 3D printed meshes, as shown in Figure 8. The implantation of PCL 3D printed meshes leads to an influx of inflammatory macrophages. However, AV hydrogel could significantly modulate this response within the acute phase [55]. More recently, we have shown that childbirth injury is associated with a drastic increase in inflammatory CCR7+ cellular response in the vaginal tissue. However, this can be rapidly reversed with AV and more so with AV and eMSC injection into an anti-inflammatory microenvironment [54]. Therefore, the growing body of evidence suggests that AV-based scaffolds have a significant potential in TE owing to their capacity to influence various cellular processes in the body.

The modulation of inflammatory responses is critical to even a skin burn healing process, whereby AV therapeutics promote angiogenesis and re-epithelialisation through reduced IL-1β. This is particularly important for the success of cell-based therapies given that IL-1β in the local microenvironment reduces the therapeutic efficiency of stem cells [104]. Given the success of implanted constructs is determined by the innate and adaptive immune responses, it is important to engineer such devices with molecules that can overcome the challenge of FBR. The rich bioactive components of AV provide an opportunity to design biomaterial constructs and effectively modulate such immune responses and enhance the efficiency of therapeutic cells. Therefore, incorporating AV and its components is an attractive strategy for overcoming the immunobiological hurdles in TE.

## 7. Conclusions

TE aims to address the complex challenges in healthcare and involves several facets in engineering, chemistry, cell biology and immunology to effectively regenerate tissues. AV and its components have an intrinsic healing, anti-inflammatory and anti-microbial properties that can substantially improve the repair process. A growing body of evidence suggests that AV directly impacts the immune system and is thereby likely to impact the fate of implanted biomaterials and cells. Thus, it is desirable to incorporate AV in scaffolds to impart reparative properties without piquing adverse FBR from a clinical translational perspective. This presents an inter-disciplinary challenge to design AV-based systems with optimal biomechanical properties while keeping its biological influence intact. Development of such an AV-based implantable device requires engineers, clinicians, biochemists and material scientists to work synergistically to ensure clinical translation. Furthermore, to develop successful AV-based medical devices in the form of TE constructs, there is a pivotal need to understand the mechanisms involved in healing and immune regulation. Our current understanding of the interaction between AV and cells of the immune system is somewhat limited, thereby underlining the need to uncover the conundrums of underlying mechanisms. Despite the challenges and substantial need for further research, AV and its bioactive components are highly promising macromolecules for enhancing regenerative outcomes of TE and cellular therapies.

## Figures and Tables

**Figure 1 ijms-22-01708-f001:**
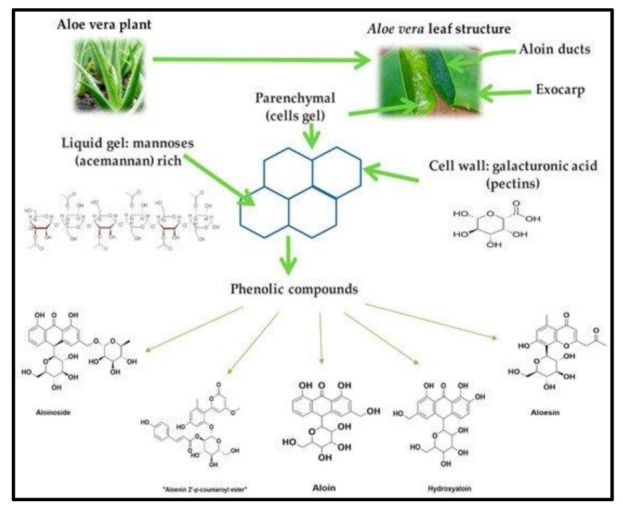
*Aloe vera* plant and chemical structure of its main components. Reproduced with permission from [12].

**Figure 2 ijms-22-01708-f002:**
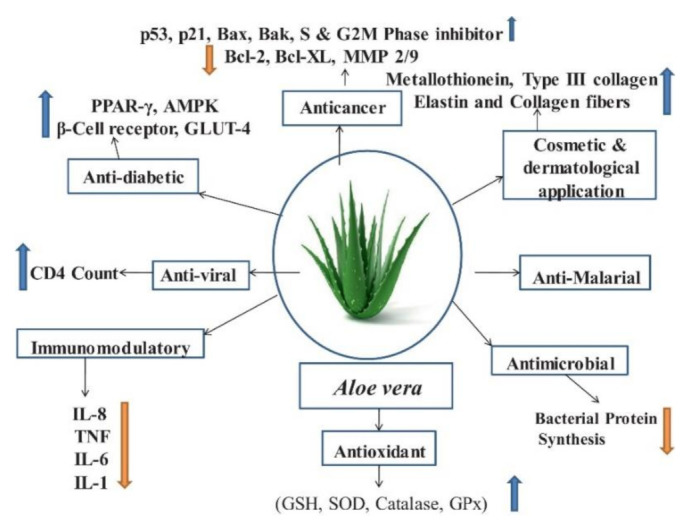
Schematic outlining the biological properties of *Aloe vera*. Reproduced with permission from [2]; copyright Elsevier, 2019.

**Figure 3 ijms-22-01708-f003:**
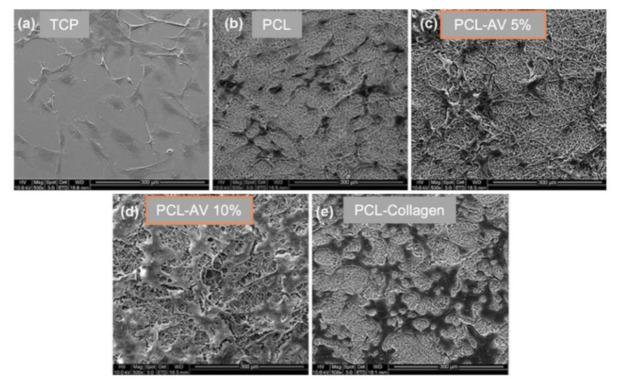
**Fibroblast interaction with *Aloe vera* (AV)-based nanofibrous scaffolds.** This figure shows different mesh structures infused with AV at 5% and 10% and collagen. The interactions of fibroblasts with the mesh surface are measured using scanning electron microscopy (SEM). The dark patches within each image are indicative of the fibroblasts. Reproduced with permission from [64], copyright Iran Polymer and Petrochemical Institute, 2014.

**Figure 4 ijms-22-01708-f004:**
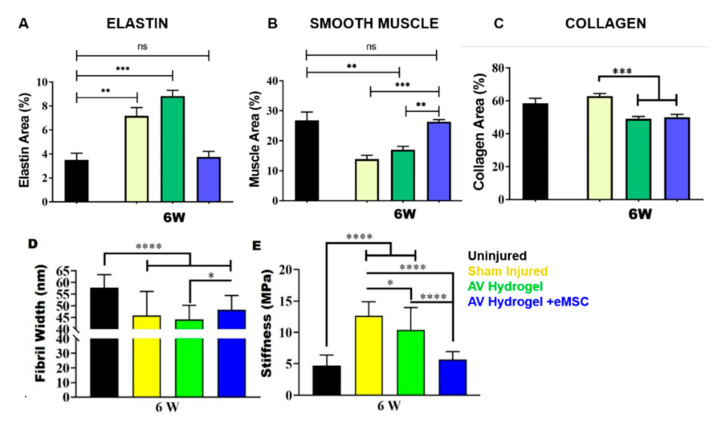
**Impact of AV hydrogel therapeutic in birth injury treatment.** The vaginal injection of AV hydrogel with and without endometrial mesenchymal stem cells (eMSCs) reverses the acute childbirth trauma as seen through vaginal tissue components such as (**A**) elastin, (**B**) smooth muscle cells and (**C**) tissue collagen. The AV-based treatment also shows nanoscopic changes in (**D**) collagen fibril width and (**E**) stiffness measured by atomic force microscopy. Statistical analysis one-way ANOVA with Tukey’s multiple comparisons test (*, *p* < 0.05, ** *p* < 0.01, *** *p* < 0.001, ****, *p* < 0.0001).Reproduced with permission from [54], copyright Elsevier, 2021.

**Figure 5 ijms-22-01708-f005:**
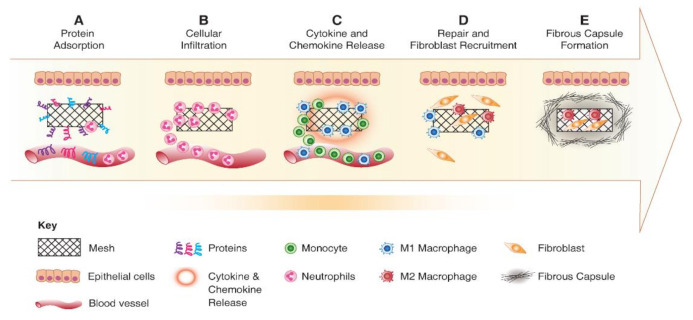
**Schematic showing the foreign body response to implanted biomaterial**. (**A**) Protein adsorption; (**B**) cellular infiltration; (**C**) chronic inflammation; (**D**) fibroblast recruitment and matrix deposition; (**E**) fibrous capsule formation. Reused with permission from [53].

**Figure 6 ijms-22-01708-f006:**
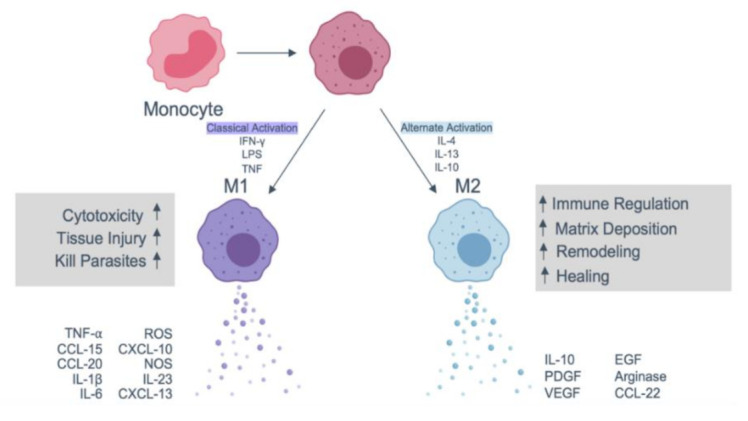
**Schematic showing macrophage polarisation**. The process is triggered during macrophage activation and polarisation into M1 and M2 subtypes and the subsequent cytokines and chemokines released by these cells. Adapted with permission from [53].

**Figure 7 ijms-22-01708-f007:**
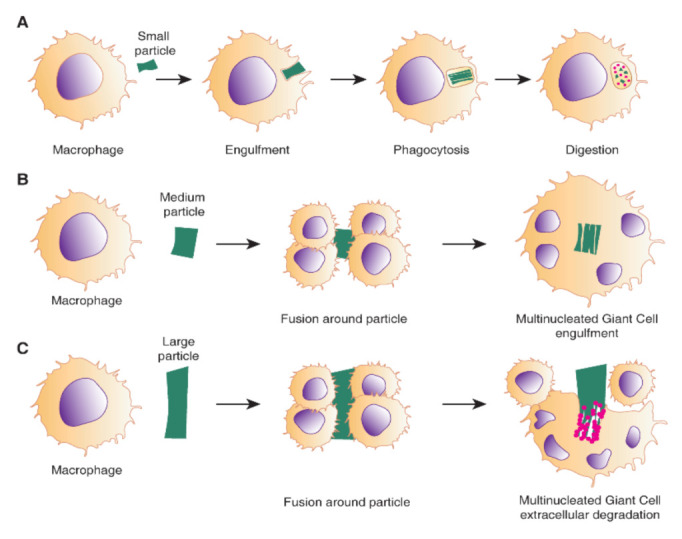
**Schematic showing formation of foreign body giant cells (FBGCs)**. Macrophages respond to foreign bodies in the host by (**A**) phagocytosis. However, when the particle is larger than a single macrophage, (**B**) they fuse to form multinucleated FBGCs around the particle, fully encapsulating it. (**C**) When the particle is much larger than an FBGC, multiple FBGCs attempt to fuse around the foreign particle to render extracellular degradation. Reproduced with permission from [53].

**Figure 8 ijms-22-01708-f008:**
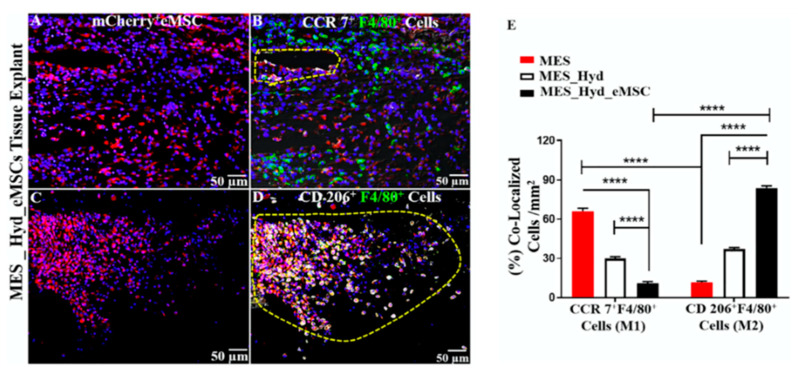
**Immune response and infiltration of host cells into melt electrospinning (MES) mesh.** (**A**–**D**) Immunofluorescence showing m-Cherry+ eMSCs labelled (pink) and recruitment of CCR7+ (M1) and CD206+ (M2) macrophages in an explant after 1 week. (**E**) Quantified chart showing co-localisation of M1/M2 macrophages. Figure shows that the presence of eMSCs was consistent with a reduction in inflammatory M1 macrophages and an increase in healing M2 macrophages. (****, *p* < 0.0001). Reproduced with permission from [55].

**Table 1 ijms-22-01708-t001:** **Components of *Aloe vera*.** Adapted with permission from [6].

Type	Compounds
Anthraquinones/ anthrones	Aloe-emodin, aloetic acid, anthranol, aloin A and B (collectively known as barbaloin), isobarbaloin, emodin, ester of cinnamic acid
Carbohydrates	Pure mannan, acetylated mannan, acetylated glucomannan, glucogalactomannan, galactan, pectic substance, arabinogalactan, galactoglucoarabinomannan, galactogalacturan, xylan, cellulose
Enzymes	Alkaline phosphatase, amylase, carboxypeptidase, carboxylase, catalase, cyclooxidase, phosphoenolpyruvate, cyclooxygenase, superoxide dismutase, lipase, oxidase
Inorganic compounds	Calcium, chlorine, phosphorous, chromium, copper, magnesium, iron, manganese, potassium, sodium, zinc
Non-essential and essential amino acids	Alanine, arginine, aspartic acid, glutamic acid, glycine, histidine, hydroxyproline, isoleucine, leucine, lysine, methionine, proline, threonine, tyrosine, valine, phenylalanine
Proteins	Lectins, lectin-like substance
Saccharides	Mannose, glucose, l-rhamnose, aldopentose
Vitamins	B1, B2, B6, C, β-carotene, choline, folic acid, α-tocopherol
Miscellaneous	Arachidonic acid, γ-linolenic acid, potassium sorbate, steroids (campestrol, cholesterol, β-sitosterol), triglycerides, triterpenoid, gibberellin, lignins, salicylic acid, uric acid

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
