# Peer review of "Immunobiology and Application of Aloe vera-Based Scaffolds in Tissue Engineering"

_ijms, 2021, doi:10.3390/ijms22041708_

Round 1

Reviewer 1 Report

Dear authors,

The manuscript “Immunobiology and Application of Aloe Vera based Scaffolds in Tissue Engineering” is very interesting, well organised and presented. It fits perfectly with some of the latest research trends on functionalised biomaterials for Tissue Engineering, providing several hints in this research area. Therefore, I strongly recommended the publication of this manuscript, after “minor revisions”.  Most of my revisions are reported and highlighted in yellow in the manuscript pdf version (please see the attached file). Specifically, you will find expressions crossed out in red with the relative correction and some reported as comments in the yellow box. Obviously, my corrections are only suggestions that I have proposed and that seemed more fitting to me while I was reading the manuscript. Furthermore, below there are some of my other comments/suggestions:

  • Page 1, line 2-3, please change “Aloe Vera based Scaffolds” in “Aloe Vera-based Scaffolds” in the whole manuscript, where it is intended that the scaffolds or hydrogels are based on AV
  • Page 3, please could the authors specify what the following expressions refer to:
    • line 85, “(0.25%)
    • line 89, “(SAHA)
    • line 106-107 “attributes of the gel
    • lines 89-91 please check the sentence “Although not fully understood, the structure of carbohydrates in AV gel show linkage and optical rotation analysis” by changing “show” with “has been demonstrated by” or changing with “shows linkage and optical rotation properties”
  • Page 7, lines 242-245 “Tissue engineering (TE), a branch of regenerative medicine, is one the most exciting and rapidly growing field of biomedical science. Tissue engineering is combination of biological, material and stem cells science which together develop an optimal regeneration of damaged tissue”, please could the authors provide a better introduction about tissue engineering? Please, could the authors specify what do they means with “t” in the line 284?
  1. Page 8, line 323 please could be the authors more specific in the expression “the impact of AV..”, explaining which property or effect do they refer to?
  2. Page 9, line 330 what do the authors mean with the term “augment”?

Please, check and correct the layout of the Figure 4 caption (lines 345-349), which is not directly below the figure.

  1. Page 10, line 401 the sense of “identified” was not clear for me
  2. Some references are repeated. Would the authors be so patient to check all the references and their relative numerical correspondences in the manuscript? Here are the ones I noticed during the manuscript revision:

[1] and [8] Gao, Y.; Kuok, K.I., et al. Biomedical applications of Aloe vera. Crit Rev Food Sci Nutr 2019, 59, S244-s256, 558. doi: 10.1080/10408398.2018.1496320

[26] and [28] Zhang, L.; Tizard, I.R. Activation of a mouse macrophage cell line by acemannan: The major carbohydrate fraction from Aloe vera gel. Immunopharmacology 1996, 35, 119-128. doi: 10.1016/S0162-3109(96)00135-X

[61] and [62] Paul, K.; Darzi, S., et al. 3D bioprinted endometrial stem cells on melt electrospun poly ε-caprolactone mesh for pelvic floor application promote anti-inflammatory responses in mice. Acta Biomaterialia 2019, 97, 162-176. doi: 10.1016/j.actbio.2019.08.003

[77] and [89] Budai, M.M.; Varga, A., et al. Aloe vera downregulates LPS-induced inflammatory cytokine production and expression of NLRP3 inflammasome in human macrophages. Molecular immunology 2013, 56, 471-479. doi: 10.1016/j.mo-716 limm.2013.05.005

[90] and [91] Habeeb, F.; Stables, G., et al. The inner gel component of Aloe vera suppresses bacterial-induced pro-inflammatory cytokines from human immune cells. Methods 2007, 42, 388-393. doi: 10.1016/j.ymeth.2007.03.005.

  1. Finally, I would like to invite the authors to give a whole check and revision of the English (in order to remove some mistakes) and the layout of the Figures, captions and references according to the guidelines of the Journal

Author Response

Response to reviewer 1 comments
We thank the reviewer for supporting comments. Modifications as per reviewer comments are highlighted in yellow in the revised manuscript.

Page 1, line 2-3, please change “Aloe Vera based Scaffolds” in “Aloe Vera-based Scaffolds” in the whole manuscript, where it is intended that the scaffolds or hydrogels are based on AV

Response: We thank the reviewer for their suggestion. We have changed the phrase and the suggested new text (highlighted yellow) on page 1 Line 2, page 1 line 20 and 21, page 2 line 62, page 7 line 250, page 8 line 299 and 324, page 9 line 339 and 351, page 12 line 460,  page 14 line 522, 545 and 546 in the revised manuscript.

Page 3, please could the authors specify what the following

line 85, “(0.25%)”

Response: We re-wrote the sentence and it is highlighted in yellow on page 3 line 83-86. “AV gel mainly contains over 99% water. Beside water, other material components account for about 0.5-1% of AV gel. Of these, the carbohydrate containing fractions (~0.25% of AV gel) is the major fraction which has been reported to vary from 25-50% of solid precipitates”

line 89, “(SAHA)”

Response: We apologise for this error. We removed “SAHA” and added the related reference (highlighted din yellow) on page 3 line 89.

line 106-107 “attributes of the gel”

Response: We changed the word “attribute” to “properties”. This is highlighted on page 3 line 107.

lines 89-91 please check the sentence “Although not fully understood, the structure of carbohydrates in AV gel show linkage and optical rotation analysis” by changing “show” with “has been demonstrated by” or changing with “shows linkage and optical rotation properties”.

Response: we appreciate reviewer comment. we changed “show” to “has been demonstrated”. It is highlighted on page 3 line 89-91.

Page 7, lines 242-245 “Tissue engineering (TE), a branch of regenerative medicine, is one the most exciting and rapidly growing field of biomedical science. Tissue engineering is combination of biological, material and stem cells science which together develop an optimal regeneration of damaged tissue”, please could the authors provide a better introduction about tissue engineering?

Response:

Thank you for the reviewer’s suggestion. We modified the introduction as “Tissue engineering is a thriving field of regenerative medicine that aims to develop biocompatible substitutes that restore or improve damaged tissue function. Tissue engineering involves scientists with different expertise to enhance people's quality of life with critical health issues” on page 7 line 238-252.

Please, could the authors specify what do they means with “t” in the line 284?

Response: We apologize for this typo. We removed “t” on page 7 line 284.

Page 8, line 323 please could be the authors more specific in the expression “the impact of AV..”, explaining which property or effect do they refer to?

Response: This refers to the immunological implications of the AV. We modified the sentence and it is highlighted on page 8 line 309-311. “Yet, the impact of blending AV into the polymeric meshes has not yet been fully explored. In particular, the in vivo immunological implications remain unknown.”

Page 9, line 330 what do the authors mean with the term “augment”?

Response: This sentence has been modified and it is highlighted in yellow on page 9 line 317-318.

“Combining hydrogels with electrospun meshes can augment the tissue integration properties of the overall TE construct.”

Please, check and correct the layout of the Figure 4 caption (lines 345-349), which is not directly below the figure.

Response: The figure layout is corrected and the caption is below figure on page 9.

Page 10, line 401 the sense of “identified” was not clear for me

Response: We modified the sentence and it is highlighted in yellow on page 10 line 388-391. “Acute and chronic phases of FBR follow the formation of the provisional matrix. The intensity of FBR is primarily determined by the composition and design properties of implanted scaffold, and the extent of the damage incurred”.

Some references are repeated. Would the authors be so patient to check all the references and their relative numerical correspondences in the manuscript? Here are the ones I noticed during the manuscript revision:

[1] and [8] Gao, Y.; Kuok, K.I., et al. Biomedical applications of Aloe vera. Crit Rev Food Sci Nutr 201959, S244-s256, 558. doi: 10.1080/10408398.2018.1496320

[26] and [28] Zhang, L.; Tizard, I.R. Activation of a mouse macrophage cell line by acemannan: The major carbohydrate fraction from Aloe vera gel. Immunopharmacology 199635, 119-128. doi: 10.1016/S0162-3109(96)00135-X

[61] and [62] Paul, K.; Darzi, S., et al. 3D bioprinted endometrial stem cells on melt electrospun poly ε-caprolactone mesh for pelvic floor application promote anti-inflammatory responses in mice. Acta Biomaterialia 201997, 162-176. doi: 10.1016/j.actbio.2019.08.003

[77] and [89] Budai, M.M.; Varga, A., et al. Aloe vera downregulates LPS-induced inflammatory cytokine production and expression of NLRP3 inflammasome in human macrophages. Molecular immunology 201356, 471-479. doi: 10.1016/j.mo-716 limm.2013.05.005

[90] and [91] Habeeb, F.; Stables, G., et al. The inner gel component of Aloe vera suppresses bacterial-induced pro-inflammatory cytokines from human immune cells. Methods 200742, 388-393. doi: 10.1016/j.ymeth.2007.03.005.

Response:  We apologise for this error. Repeated references were removed and updated.

 Finally, I would like to invite the authors to give a whole check and revision of the English (in order to remove some mistakes) and the layout of the Figures, captions and references according to the guidelines of the Journal

Response:  We have now edited the grammar of the manuscript using Grammarly premium and modified it correctly. We have also adhered to the journal guidelines for the figure and caption and used the MDPI endnote plug in for referencing style as provided on the MDPI website.

Reviewer 2 Report

The Authors present in this Review the growing scientific evidence on the advantage the use of Aloe Vera (AV) based scaffold in tissue engineering in both hydrogel and membranous form, underlining that implanted AV functionalized scaffolds strike an appropriate foreign body response to be well accepted in the body without complications. Herein, it is highlight the key players that regulate the macrophage mediated body immune response and are emphasise the immunomodulatory effects incurred by AV on innate and adaptive system. In line with AV multiple beneficial components described, it is discussed the importance of delving deeper into uncovering the mechanism of AV action and thereby improve material design strategies for better tissue engineering constructs for biomedical applications.

The Manuscript is well organized and well written. It provides solid information about composition, properties and beneficial effects of AV in the healing, anti-inflammatory and anti-microbial properties that can substantially improve the repair and regeneration.

Appropriate References are cited.

In light of the increasing beneficial evidences of direct AV effects on the immune system and of the influence on the fate of biomaterials and cells illustrate in the manuscript, accordingly to Authors Conclusion, there is an urgent need to deepen complex molecular mechanisms involved in healing to develop AV-based proper medical devices to be used Tissue Engineering.

No revision are required.

Author Response

We thank you the reviewer for positive comments and accepting our manuscript.

Reviewer 3 Report

The authors proposed covering the mechanism of action of aloe vera and strategies to design material using aloe vera for biomedical applications. An extensive description of the papers on aloe vera was made, but it was missing a critical analysis from the authors, and therefore, its scientific relevance is not so clear.

Some specific comments are described below:

  • The term “membranous form” is not an adequate form. Please remove it along with the manuscript.
  • The connection between the paragraphs along the manuscript is not so clear in several parts. The authors tried to cover many aspects of aloe vera, and maybe it should be more focused on one or two aspects to demonstrate its originality.
  • Section 2 should be revised. It seems a compilation of the papers on aloe vera already published in the literature.
  • Page 7, section 4, the general concept of tissue engineering used in this section should be summarized. Moreover, appropriate references should be added to this section to support the discussion.
  • Page 7, “ The natural polymers such as AV”…please explain.
  • Page 8, From a translation perspective, it is important to not only design effective scaffolds for tissue repair,…..that may impact its degradation”. Please explain the relevance of this statement in the context of the use of aloe vera.
  • Page 19, section 5 described a general concept of foreign body response to implanted biomaterials without any reference about aloe vera. This section's relevance is not clear, and it should be summarized, and its significance on aloe vera explained.
  • Page 12, considering the authors' papers about the immunobiology of aloe vera described and the papers published in the literature, this section should be more explored. A better discussion should be made.  
  • Page 14, the conclusions section was too poor, and it should be revised to demonstrate what is expected for the use of aloe vera in tissue engineering, challenges, and perspectives.
  • Page 15, many duplicate references were found. Please revise it.

Author Response

We thank the reviewer for supporting comments. Modifications as per reviewer comments are highlighted in yellow in the revised manuscript

Response to Reviewer 3 Comments:

The authors proposed covering the mechanism of action of aloe vera and strategies to design material using aloe vera for biomedical applications. An extensive description of the papers on aloe vera was made, but it was missing a critical analysis from the authors, and therefore, its scientific relevance is not so clear.

Response:  Aloe Vera (AV), as described in the manuscript, is a plant with many medicinal properties. The study of its application involves cross disciplinary approaches in biochemistry, cell biology and material science. Moreover, it has recently found application in tissue engineering owing to its healing properties. As described in the abstract and introduction of the manuscript, we provide a comprehensive summary regarding the constituents of AV and its immunobiology. Our paper provides insights into the immunological reaction triggered by AV in the body. This is highly relevant and important for cross disciplinary researchers who wish to use bioactive natural biomaterials such as AV for design of tissue engineering construct. Yet another relevance of this manuscript lies in the fact that in tissue engineering, immune regulation is an important aspect. Implantable AV based scaffolds present itself as an animal component free construct for influencing cells, thus highly valuable from a clinical translational perspective. We have discussed our perspective on the importance and application of AV in each section as well as in the conclusion.

The term “membranous form” is not an adequate form. Please remove it along with the manuscript

Response:  We removed “membranous form” on page 1 line 21-22 (highlighted in yellow)

The connection between the paragraphs along the manuscript is not so clear in several parts. The authors tried to cover many aspects of aloe vera, and maybe it should be more focused on one or two aspects to demonstrate its originality.

Response:  Tissue engineering is a cross disciplinary research which requires understanding of biochemistry of components, principles of material fabrication, cellular interaction biology as well as immune response. While there are many other papers that have focussed on individual aspect of AV, our manuscript brings together the multiple aspects which are essential for using AV in form of scaffolds for tissue engineering. This review aims to provide inter disciplinary researchers an overview of the multi-modal considerations in designing AV based scaffold. Hence, we believe the diversity is a strength and important aspect of our manuscript.

Section 2 should be revised. It seems a compilation of the papers on aloe vera already published in the literature.

Response:  We have now revised some parts of this section to improve its readability. We have also added new references. The goal of this section is to summarize the components of AV. This information will be essential for researchers who have a clinical background and do not know much about biochemistry.

Page 7, section 4, the general concept of tissue engineering used in this section should be summarized. Moreover, appropriate references should be added to this section to support the discussion.

Response:  We thank the reviewer for their comment. This section was shortened and new references were added. The paragraph is highlighted on page 7 line 238 to 252.

Page 7, “The natural polymers such as AV”…please explain.

Response:  We have now revised this sentence. We replaced “natural polymers” with “natural biomaterials” on page 6 line 245,  7 line 256,273 and 275.

Page 8, From a translation perspective, it is important to not only design effective scaffolds for tissue repair,…..that may impact its degradation”. Please explain the relevance of this statement in the context of the use of aloe vera.

Response:  The aim of incorporating AV with polymers to design tissue engineering scaffold is to impart the healing properties of AV into the scaffold. When such a scaffold in implanted in the body, the interactions with immune cells and other host cells will release cytokines and growth factors. This mechanism of action that drives the healing process is important to characterise. Many enzymes involved in tissue remodelling process such as MMPs and TIMPs are likely to have an impact on the degradation of scaffolds. The degradation process in turn will further impact the healing and immune response. Thus, it is very important to study which factors are released, what effect it will have on the design of the scaffold and how it will impact degradation of scaffold. The important elements of this is described in section 5.

Page 19, section 5 described a general concept of foreign body response to implanted biomaterials without any reference about aloe vera. This section's relevance is not clear, and it should be summarized, and its significance on aloe vera explained.

Response:  The aim of this section is to introduce researchers without immunology background to the concept of foreign body response. The immunobiology of AV is described in the following section- section 6. We added a short paragraph to connect the part “foreign body response” to “immunobiology of AV”. It is highlighted on page 12 line 456-459.

Page 12, considering the authors' papers about the immunobiology of aloe vera described and the papers published in the literature, this section should be more explored. A better discussion should be made.  

Response: We appreciate reviewer’s comment and agree that this is an interesting aspect. However, there is not much known about the immune response to aloe vera. Our manuscript aims to highlight the importance of exploring more in this area. As per reviewer’s suggestion, we have added few more studies on AV modulatory effects on wound healing response. It is highlighted on page 13 line 506-513.

Page 14, the conclusions section was too poor, and it should be revised to demonstrate what is expected for the use of aloe vera in tissue engineering, challenges, and perspectives.

Response:  We have now modified the conclusion to add our perspective on the recommended aspects as per the reviewer.

Page 15, many duplicate references were found. Please revise it.

Response:  We have now rectified this error and revised our manuscript.

Round 2

Reviewer 3 Report

It seems that the authors made an effort to improve the quality of the manuscript. However, some minor corrections should be made.

  • The reviewers highlighted again that the term “membranous form” is not an acceptable form and it should be eliminated in the whole manuscript. This term still appears on page 7, line 253; line 260; line 279
  • Page 7, line 292, the term “ scaffolded” is not appropriate, and it should be corrected.
  • Some references appeared without pages as e.g., ref 6, 12, 69

Author Response

Reviewer 3:

It seems that the authors made an effort to improve the quality of the manuscript. However, some minor corrections should be made.

Author Response: We thank the reviewer in taking the time and effort to carefully review our manuscript. The points listed by reviewer has definitely helped us improve our manuscript in the past and present review process. We have further improved the English grammar of our manuscript and highlighted them in blue. We have responded to each point raised by reviewer below. 

1.The reviewers highlighted again that the term “membranous form” is not an acceptable form and it should be eliminated in the whole manuscript. This term still appears on page 7, line 253; line 260; line 27

Author Response: We thank the reviewer for pointing out the 2 places we missed to edit the word. We also found 1 more place where this word appeared in our search. We have now changed them in the manuscript and highlighted in blue. 

2. Page 7, line 292, the term “ scaffolded” is not appropriate, and it should be corrected.

Author Response: We thank the reviewer for carefully pointing out this typo. We have now corrected this word and highlighted in blue.

3.Some references appeared without pages as e.g., ref 6, 12, 69

Author Response: We thank the reviewer for pointing this out. These 3 references are from MDPI open access journals which do not have page number. We have referenced as recommended and indicated in each of these MDPI journal.